# N123I mutation in the ALV-J receptor-binding domain region enhances viral replication ability by increasing the binding affinity with chNHE1

**Mengmeng Yu**[1]☯, **Yao Zhang**[1]☯, **Li Zhang**[1], **Suyan Wang**[1], **Yongzhen Liu**[1], **Zhuangzhuang Xu**[1], **Peng Liu**[1], **Yuntong Chen**[1], **Ru Guo**[1], **Lingzhai Meng**[1], **Tao Zhang**[1], **Wenrui Fan**[1], **Xiaole Qi**[1], **Li Gao**[1], **Yanping Zhang**[1], **Hongyu Cui**[1], **Yulong Gao**[1,2,3]*

**1** Avian Immunosuppressive Diseases Division, State Key Laboratory for Animal Disease Control and Prevention, Harbin Veterinary Research Institute, the Chinese Academy of Agricultural Sciences, Harbin, China, **2** Jiangsu Co-innovation Center for Prevention and Control of Important Animal Infectious Disease and Zoonoses, Yangzhou, China, **3** National Poultry Laboratory Animal Resource Center, Harbin, China

☯ These authors contributed equally to this work.
* gaoyulong@caas.cn

**Data Availability Statement:** All data are in the manuscript and/or supporting information files.

**Funding:** This work was supported by National Natural Science Foundation of China (32230105,

## Abstract

The subgroup J avian leukosis virus (ALV-J), a retrovirus, uses its gp85 protein to bind to the receptor, the chicken sodium hydrogen exchanger isoform 1 (chNHE1), facilitating viral invasion. ALV-J is the main epidemic subgroup and shows noteworthy mutations within the receptor-binding domain (RBD) region of gp85, especially in ALV-J layer strains in China. However, the implications of these mutations on viral replication and transmission remain elusive. In this study, the ALV-J layer strain JL08CH3-1 exhibited a more robust replication ability than the prototype strain HPRS103, which is related to variations in the gp85 protein. Notably, the gp85 of JL08CH3-1 demonstrated a heightened binding capacity to chNHE1 compared to HPRS103-gp85 binding. Furthermore, we showed that the specific N123I mutation within gp85 contributed to the enhanced binding capacity of the gp85 protein to chNHE1. Structural analysis indicated that the N123I mutation primarily enhanced the stability of gp85, expanded the interaction interface, and increased the number of hydrogen bonds at the interaction interface to increase the binding capacity between gp85 and chNHE1. We found that the N123I mutation not only improved the viral replication ability of ALV-J but also promoted viral shedding *in vivo*. These comprehensive data underscore the notion that the N123I mutation increases receptor binding and intensifies viral replication.

## Author summary

Attachment is a critical initial step in retroviral infections. The RBD of gp85 in ALV-J is critical for binding to the cell membrane receptor, chNHE1. This investigation establishes a strong link between the heightened replication ability of the layer strain JL08CH3-1 and the enhanced binding capacity exhibited by gp85 toward chNHE1 compared to the

31872482) (YG), Heilongjiang Touyan Innovation Team Program (YG), Heilongjiang Provincial Natural Science Foundation of China (YQ2023C029) (SW), and China Agriculture Research System (CARS-41) (YG). The funder had no role in study design, data collection and analysis, decision to publish, or preparation of the manuscript."

**Competing interests:** The authors have declared that no competing interests exist.

prototype strain HPRS103. We found for the first time that the N123I mutation within the RBD region of gp85 increased the binding capacity between gp85 and chNHE1 and enhanced the replication ability of ALV-J. Our results provide important information on the implications of mutations within the RBD region of gp85 and highlight the importance of monitoring variations in the RBD region of ALV-J to prevent future outbreaks.

## Introduction

Avian leukosis viruses (ALVs) are a group of avian retroviruses that cause tumors in birds [1]. ALVs in chickens can be classified into seven subgroups (A, B, C, D, E, J, and K) according to the host range, envelope properties, and cross-reactivity with neutralizing antibodies [2,3]. ALV-E is an endogenous virus with minimal or no pathogenicity [4,5]. ALV-A and ALV-B are prominent pathogenic strains that induce lymphoid leukosis and sarcoma [6,7]. In contrast, ALV-C and ALV-D infections have been reported infrequently. ALV-K primarily induces gliomas [2]. However, ALV-J infection is notable for its increased transmission and pathogenicity, which poses formidable challenges to its control and eradication [8–10].

In China, ALV-J was first detected in commercial broiler chickens in 1999 [11], followed by scattered reports of infection in broiler and local chickens [12–15]. However, no field cases of ALV-J infection or tumors in layer chickens were reported until 2004 [16]. From 2008 to 2009, ALV-J caused a large-scale epidemic in layer chickens in China, leading to the occurrence of hemangiomas in multiple-layer chicken strains [16,17]. Molecular epidemiologic investigations have shown that most gp85 genes in ALV-J layer strains formed an independent branch, and their homology with the ALV-J prototype strain HPRS103 was below 90.6% [18–20]. Some researchers have also found that ALV-J layer strains show higher replication ability in cells and induce higher expression levels of tumor-related genes in infected cells than in the broiler strain [21]. However, further research is required to investigate whether the enhanced replication ability of ALV-J layer strains is associated with genetic variation in gp85.

Attachment is the most crucial step in the infection process of retroviruses [22]. ALV-J enters cells primarily by recognizing the chicken sodium hydrogen exchanger isoform 1 (chNHE1) through the viral envelope protein (Env) [23,24]. ALV-J Env is a typical type I transmembrane protein that is assembled into a trimer at the endoplasmic reticulum, glycosylated in the Golgi, and subsequently cleaved into surface glycoprotein SU (gp85) and transmembrane TM (gp37) units. Gp85 is located on the surface of virions, which determines the host range [25–27]. The receptor-binding domain (RBD) of gp85 is a critical region that specifically binds to receptors and is the most variable region [28,29]. The previous molecular epidemiological investigation showed that compared to ALV-J broiler strains, ALV-J layer strains exhibited 13 regular amino acid mutations within gp85, 10 specifically located within the RBD region [30]. Numerous studies have shown that amino acid mutations in the RBD region may directly enhance the binding capacity of Env proteins to receptors to promote viral replication. For example, in severe acute respiratory syndrome coronavirus 2 (SARS-CoV-2), simian-human immunodeficiency virus (SHIV), and feline panleukopenia virus (FPV), amino acid mutations in RBD strongly influence viral infectivity and transmission [31–34].

Therefore, to investigate the impact of amino acid mutations in the RBD region of gp85 on the replication of ALV-J layer strains and its possible mechanisms, we initially examined the replication capability and compared the gp85 protein-binding capacity of the layer strain JL08CH3-1 and the prototype strain HPRS103. Subsequently, we constructed and expressed mutant gp85 proteins and HPRS103-gp85, evaluated their binding capacity to the receptor

chNHE1, and analyzed the possible mechanisms from a structural perspective. Our results demonstrate that the N123I mutation emerged as a pivotal factor that improves gp85 receptor binding capacity by enhancing protein stability, expanding the interaction interface, and amplifying hydrogen bond formation within the interface, improving viral replication.

## Results

### The replication advantage of the layer strain JL08CH3-1 is related to gp85

To assess the replication ability of the ALV-J layer strains, we selected the JL08CH3-1 layer strain and the prototype strain HPRS103 for *in vitro* replication experiments. Both viruses were inoculated into DF1 cells at a multiplicity of infection (MOI) of 0.01. Analysis of the samples from 1 to 7 days post-inoculation (dpi) revealed that JL08CH3-1 had higher viral titers at 3–7 dpi than HPRS103. JL08CH3-1 achieved a maximum titer of 4.69 $\log_{10}$ 50% tissue culture infective dose (TCID$_{50}$/mL), which is approximately 10 times higher than that of 3.67 $\log_{10}$ TCID$_{50}$/mL of HPRS103 (Fig 1A). As with viral titers, the reverse transcriptase (RT) assay indicated that JL08CH3-1 had significantly higher RT activity than HPRS103 at 3–7 dpi. Furthermore, RT activity of JL08CH3-1 achieved a peak titer of approximately 40.25 ng/mL, which is approximately 1.5-fold higher than HPRS103 (26.64 ng/mL) (Fig 1B). These findings indicate that layer strains have a stronger replication ability than broiler strains.

Specific binding of the gp85 protein of ALV to its receptors is a crucial step in the successful ALV infection. Compared to broiler strains, layer strains exhibited specific amino acid substitutions within the central region of the gp85 subunit. To determine whether replication differences between the layer strain JL08CH3-1 and prototype strain HPRS103 stemmed from gp85 disparities, generation of rHPRS103 (wild-type HPRS103) and rHPRS/JL3-1 (gp85 on the HPRS103 backbone replaced with the gp85 of JL08CH3-1) using reverse genetics on DF1 cells assessed using Western blotting, the result showed that rHPRS/JL3-1 and rHPRS103 were recognized by the p27 monoclonal antibody 2E5 (Fig 1C) [35]. Furthermore, indirect immunofluorescence (IFA) results also indicated the recognition of rHPRS/JL3-1 and rHPRS-103 by 4A3 (monoclonal antibodies specific for ALV-J gp85) [35], accompanied by green fluorescence (Fig 1D). These results confirm the successful rescue of rHPRS/JL3-1 and rHPRS103. When 0.01 MOI of the two viruses was inoculated into DF1 cells, the rHPRS/JL3-1 virus replicated more efficiently than the rHPRS103 virus from 3 to 7 dpi, and the titer of rHPRS/JL3-1 reached its peak (4.65 $\log_{10}$ TCID$_{50}$/mL) at 6 dpi, which was approximately 9 times higher than rHPRS103 (3.69 $\log_{10}$ TCID$_{50}$/mL) (Fig 1E). Consistent with the results of viral titers, the RT assay showed that the RT activity of rHPRS/JL3-1 was significantly higher than rHPRS103 from 3–7 dpi. Moreover, the RT activity of rHPRS/JL3-1 reached a peak titer of 37.95 ng/mL at 6 dpi, whereas the peak titer of rHPRS03 was only 26.81 ng/mL (Fig 1F). These results indicate that layer strains have a stronger replication ability and the replication advantage of layer strain JL08CH3-1 is related to gp85.

### The binding capacity of JL08CH3-1-gp85 to chNHE1 exceeds that of HPRS103-gp85 to chNHE1

The capacity of ALV-J gp85 to bind to the cell receptor, chNHE1, may directly affect the level of viral invasion of host cells. Evaluation of the protein-cell binding assay of the binding capacity of JL08CH3-1-gp85 and HPRS103-gp85 proteins to chNHE1-expressing 293T cells showed that the binding rate of the two proteins to the chNHE1 receptor increased in a dose-dependent manner, however, the difference in the binding rate between the two proteins to the receptor gradually increased as the protein concentration decreased (Fig 2A). When the

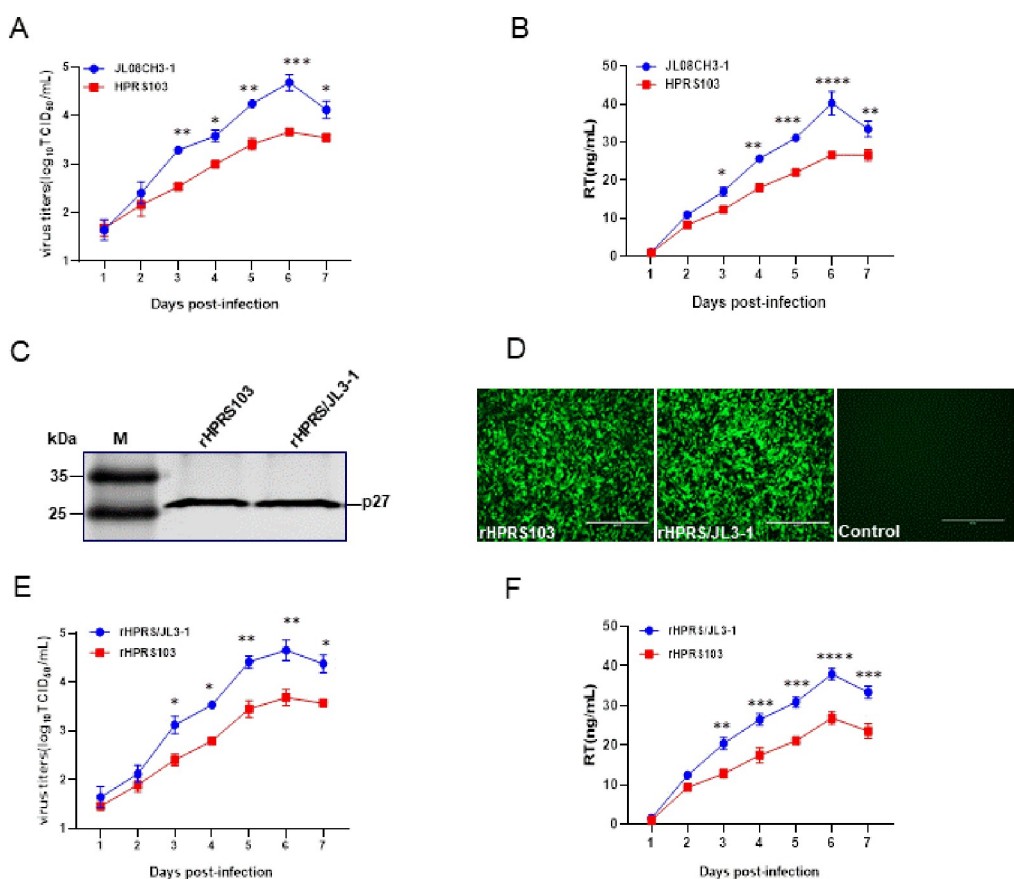

**Fig 1. Analysis of the impact of gp85 on viral replication.** (A) Virus growth kinetics. DF1 cells are infected with the layer strain JL08CH3-1 and prototype strain HPRS103 at an MOI of 0.01, which are harvested and quantified using a TCID$_{50}$ assay at 1, 2, 3, 4, 5, 6, and 7 dpi. (B) RT assay. DF1 cells are infected with the layer strain JL08CH3-1 or HPRS103 at an MOI of 0.01, which are harvested and quantified using a colorimetric reverse transcriptase assay at 1, 2, 3, 4, 5, 6, and 7 dpi. (C) Western blotting. The recombinant viruses rHPRS/JL3-1 and rHPRS103 are detected with p27 monoclonal antibody 2E5. (D) IFA assay. DF-1 cells are infected with the recombinant viruses rHPRS/JL3-1 and rHPRS103 for 72 h, and then detected with the 4A3 monoclonal antibody and analyzed using a fluorescence microscope. Scale bar: 400 μm. (E) Virus growth kinetics. DF1 cells are infected with 0.01 MOI rHPRS/JL3-1 and rHPRS103, which are harvested and quantified using a TCID$_{50}$ assay at 1, 2, 3, 4, 5, 6, and 7 dpi. (F) RT assay. DF1 cells are infected with 0.01 MOI rHPRS/JL3-1 or rHPRS103, which are harvested and quantified using a colorimetric reverse transcriptase assay at 1, 2, 3, 4, 5, 6, and 7 dpi. dpi, days post-inoculation; IFA, immunofluorescence; MOI, multiplicity of infection; TCID$_{50}$, 50% infective dose of tissue culture.

protein concentrations were 300 ng/μL, 200 ng/μL, and 50 ng/μL, the JL08CH3-1-gp85 exhibited binding rates with chNHE1 of 87.9%, 85%, and 58.5%, respectively, higher than those of HPRS103-gp85 binding cells, which were 70%, 59.2%, and 9.7%, respectively. Furthermore, the findings from the RT-qPCR and fluorescence-activated cell sorting (FACS) experiments indicate that rHPRS/JL3-1 exhibits approximately 7.96-fold and 1.29-fold greater binding capacity to chNHE1 compared to rHPRS103 in the virus-cell binding capacity experiments (Fig 2B and 2C). Our previous study demonstrated that the first extracellular loop (chECL1) binds to ALV-J gp85 and efficiently mediates ALV-J entry into cells [23]. Therefore, we co-transfected JL08CH3-1-gp85, or HPRS103-gp85, and chECL1 plasmids into 293T cells for 48 h. The co-immunoprecipitation (Co-IP) results, shown in Fig 2D, confirmed that both JL08CH3-1-gp85 and HPRS103-gp85 are efficiently bound to chECL1. However, the binding capacity of JL08CH3-1-gp85 to chECL1 was approximately 1.3-fold greater than

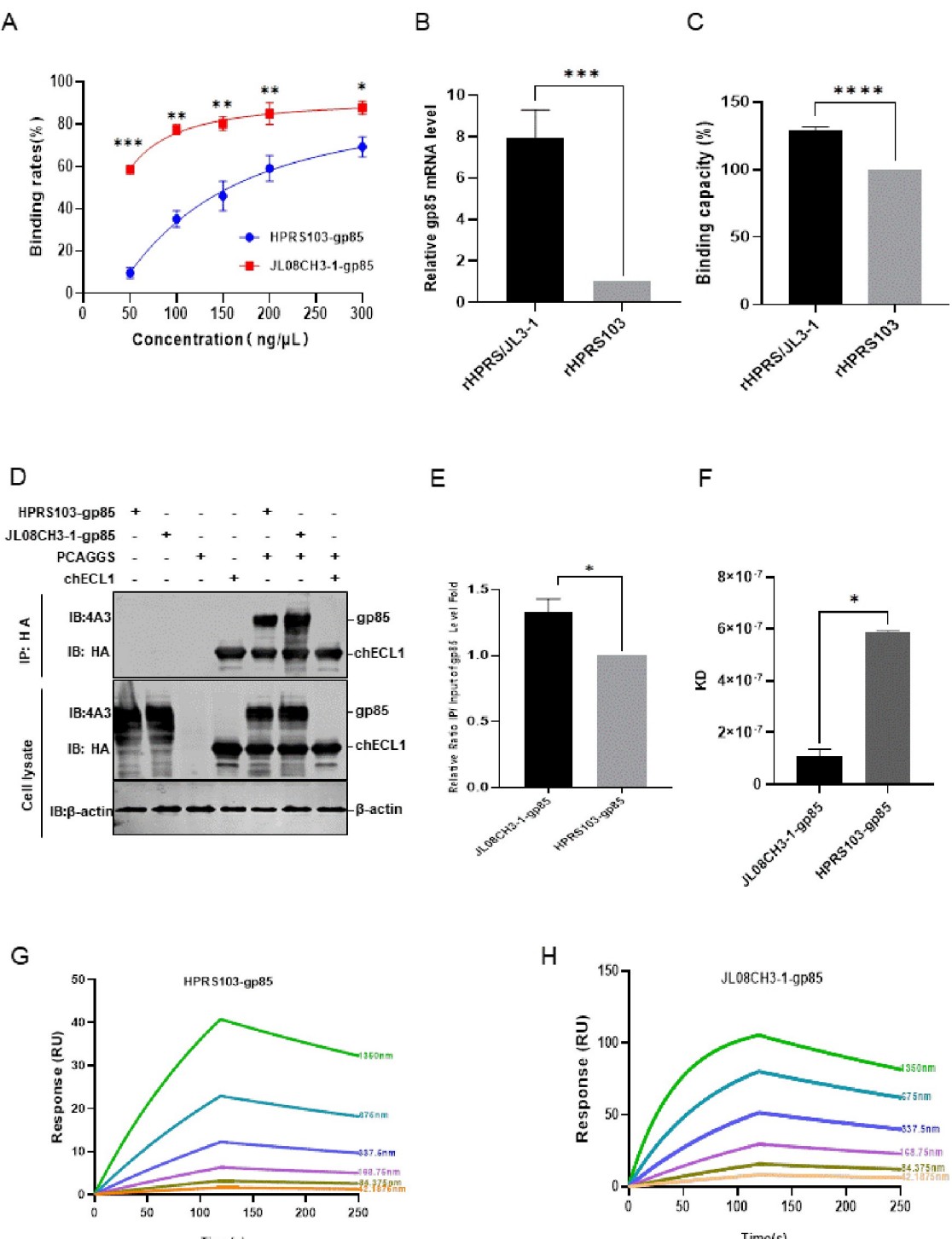

**Fig 2. Detecting the binding capacity of JL08CH3-1-gp85 and HPRS103-gp85 with the receptor chNHE1.** (A) Binding rate between different concentrations of the JL08CH3-1-gp85 and HPRS103-gp85 proteins with chNHE1 expressed on the cell membrane of transfected 293T cells. (B-C) Binding capacity between the rHPRS/JL3-1 and rHPRS103 with chNHE1 expressed on the cell membrane of transfected 293T cells is detected by RT-qPCR (B) and FCAS assays (C). (D-E) The binding capacity is detected by Co-IP assays in 293T cells. (D) 293T cells are co-transfected with chECL1 and pCAF-JL3-1-gp85 or pCAF-gp85 for 48 h. The lysates are incubated with anti-HA-agarose MAb. Lysates are detected by the 4A3 monoclonal antibody and anti-HA monoclonal antibody on Western blotting. (E) The relative intensities of gp85 are normalized to those of gp85 in the input sample. (F) Statistical summary of the KD values of JL08CH3-1-gp85 or HPRS103-gp85 protein and chECL1. (G and H) The binding kinetics of the chECL1 proteins with HPRS103-gp85 (G) or JL08CH3-1-gp85 (H) are detected using Biacore 8K (Cytiva,

Marlborough, MA, USA). chECL1 proteins are captured on the chip, and serial dilutions of gp85 then flow through the chip surface. Experiments are performed three times with similar results, and one set of representative data is displayed. Co-IP, coimmunoprecipitation; KD, equilibrium dissociation constant; chECL1, chicken first extracellular loop of chNHE1; chNHE1, chicken sodium hydrogen exchanger isoform 1.

HPRS103-gp85 with chECL1 (Fig 2E). When surface plasmon resonance (SPR) was used to investigate the binding capacities of HPRS103-gp85, JL08CH3-1-gp85 and chECL1, the equilibrium dissociation constant (KD) for the binding of HPRS103-gp85 to chECL1 was calculated to be 584 ± 9.89 nM, and the KD of the binding of JL08CH3-1-gp85 to chECL1 was 109 ± 26.3 nM (Fig 2F), indicating that the binding capacity of JL08CH3-1-gp85 to chECL1 was approximately 5.35 times greater than HPRS103-gp85 to chECL1. Further analyses of Ka and Kdis revealed that JL08CH3-1-gp85 bound faster to chECL1 (Ka $1.53 \times 10^4$ Ms vs $3.31 \times 10^3$ Ms) (Fig 2G and 2H). These results clearly illustrated that the binding capacity of JL08CH3-1-gp85 to chNHE1 was greater than HPRS103-gp85 to chNHE1.

## The N123I mutation in the ALV-J gp85 protein increases its binding capacity to the chNHE1 receptor

Compared to the amino acid sequence of HPRS103-gp85, the RBD of JL08CH3-1-gp85 exhibited 23 nonsynonymous mutations (Fig 3A). To identify the crucial amino acids responsible for improving the binding capacity between gp85 and the chNHE1 receptor, we first designed a panel of mutant gp85 proteins in which 23 amino acids from HPRS103-gp85 were replaced with the corresponding residues from JL08CH3-1-gp85, according to the approach described in our previous study [35]. Thirteen mutant gp85 plasmids were constructed and successfully expressed in 293T cells (Fig 3B). The results of the protein-cell binding assay revealed that most mutants exhibited a similar binding capacity to chNHE1 to that of HPRS103-gp85, except for the N123I/V128F mutant, which displayed a binding capacity approximately 1.3 times greater than HPRS103-gp85 to chNHE1 (Fig 3C). To further investigate which amino acid (N123I or V128F) plays a significant role in affecting the binding capacity of gp85 protein to the chNHE1, N123I and V128F single mutant plasmids were constructed and expressed in 293T cells (Fig 3D). The protein-cell binding assay results showed that the binding capacity of N123I-gp85 with chNHE1 was similar to that of the N123I/V128F mutant with chNHE1, approximately 1.3 times higher than HPRS103-gp85 with chNHE1, and V128F-gp85 exhibited a binding capacity similar to that of HPRS103-gp85 with chNHE1 (Fig 3E). The Co-IP results also confirmed that N123I-gp85 led to a roughly 1.6-fold increase in the binding of gp85 to chECL1 (Fig 3F and 3G). Furthermore, the SPR results indicated that the KD of HPRS103-gp85 binding to chECL1 was 547 ± 56.5 nM, whereas the KD of N123I-gp85 binding to chECL1 was 135 ± 17.7 nM (Fig 3H), demonstrating that the binding capacity of N123I-gp85 to chECL1 was 4.05 times greater than HPRS103-gp85 to chECL1. Further analyses of Ka and Kdis revealed that N123I-gp85 bound faster to chECL1 (Ka $1.15 \times 10^4$ Ms vs $3.73 \times 10^3$ Ms) (Fig 3I and 3J). Taken together, these results strongly suggest that N123I improves the binding capacity of ALV-J gp85 to chNHE1.

## The structural basis of N123I enhances the binding capacity of gp85 to chNHE1

To elucidate the mechanism underlying the increased binding capacity of gp85 with the N123I mutant with chNHE1, the structure of HPRS103-gp85, N123I-gp85, and chECL1 with transmembrane region (TM-chECL1) was predicted and analyzed. The results revealed that Asn123 of HPRS103-gp85 formed hydrogen bonds with Asp125 and Gly206 of HPRS103-gp85 in the

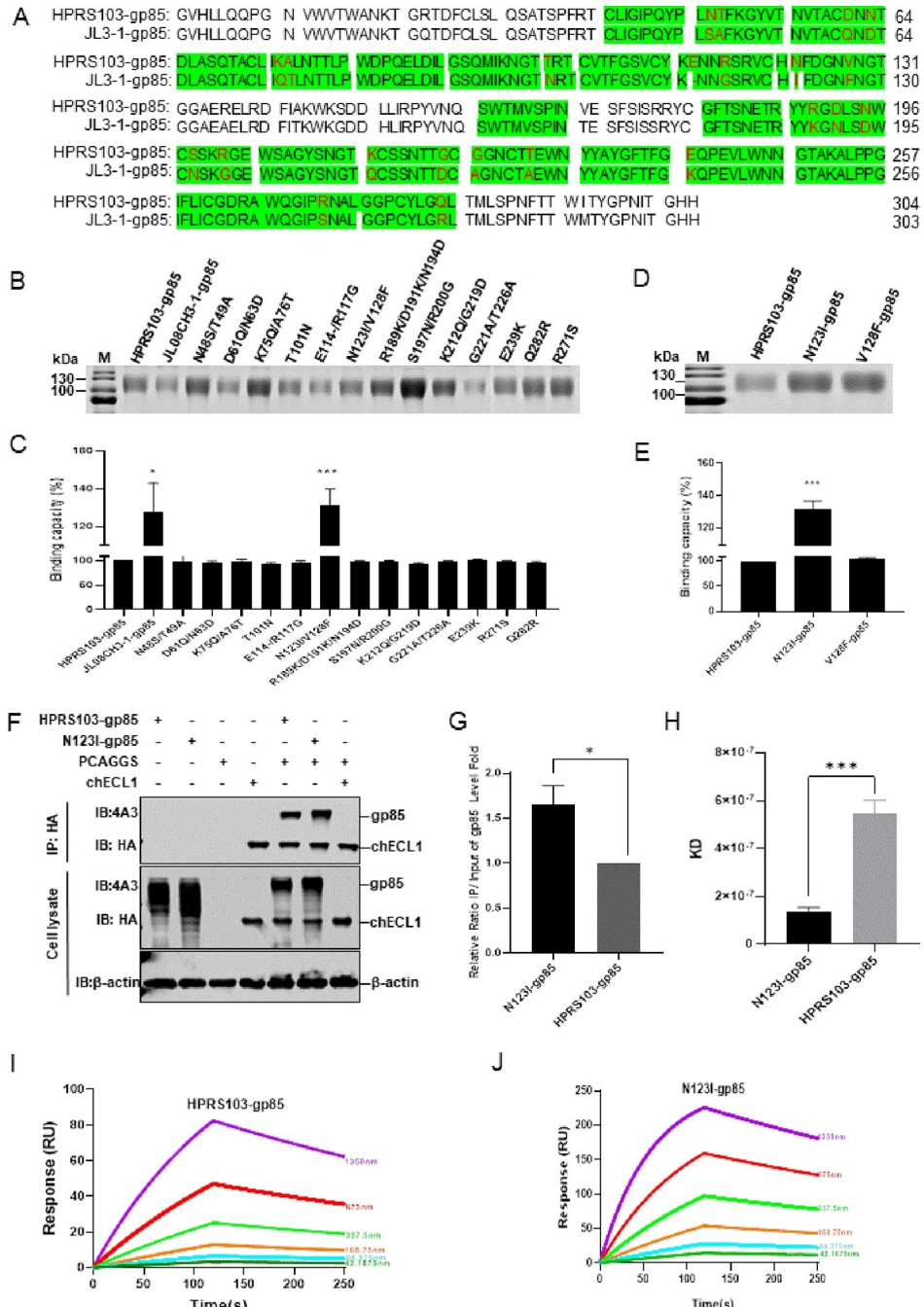

**Fig 3. Increased binding of the N123I protein to the receptor.** (A) Amino acid sequence alignment analysis for gp85 from HPRS103 and JL08CH3-1. Green shading represents the chNHE1-binding domain of ALV-J-gp85. The amino acid mutations within the RBD region are marked in red. (B) SDS-PAGE analysis of the JL08CH3-1-gp85, HPRS103-gp85, and 13 mutation gp85 proteins (according to the previous approach to construction). Fifteen plasmids containing CAG-sgp85-Fc, CAG-JL3-1-p85-Fc, and 13 mutation gp85 proteins are transfected into 293T cells for 48 h, then harvested supernatant of cells is purified by using Protein A Resin. Subsequently, the purified gp85 proteins are verified by SDS-PAGE analysis. (C) Binding capacity between the JL08CH3-1-gp85, HPRS103-gp85, and 13 mutation gp85 proteins with chNHE1 expressed on the surface of transfected 293T cells. (D) SDS-PAGE analysis of the N123I-gp85, V128F-gp85, and HPRS103-gp85 proteins. The CAG-sgp85-Fc, CAG-N123I-gp85-Fc, and CAG-V128F-gp85-Fc plasmids are transfected into 293T cells for 48 h, then harvested supernatant of cells is purified using Protein A Resin. Subsequently, the purified gp85 proteins are verified using SDS-PAGE analysis. (E) Binding capacity between the N123I-gp85, V128F-gp85, and HPRS103-gp85 proteins with chNHE1 expressed on the surface of transfected 293T cells. (F and G) The binding capacity is detected by Co-IP assays in 293T cells. (F) 293T cells are co-transfected with

chECL1 and pCAF-N123I-gp85 or pCAF-gp85 for 48 h. The lysates are incubated with anti-HA-agarose MAb. Then, the lysates are detected by 4A3 monoclonal antibody and anti-HA monoclonal antibody using Western blotting. (G) Relative intensities of gp85 are normalized to the gp85 in the input sample. (H) Statistical summary of the KD values of N123I-gp85 or HPRS103-gp85 and chECL1. (I and J) The binding kinetics of the chECL1 proteins with HPRS103-gp85(I) or N123I-gp85 (J) are detected using the Biacore 8K (Cytiva, Marlborough, MA, USA). ChECL1 protein is captured on the chip, and serial dilutions of gp85 then flow through the chip surface. Experiments are performed three times, revealing similar results, and one set of representative data is displayed. The error bars represent the SD. *, $P < 0.05$; **, $P < 0.01$; *** and ****, $P < 0.001$; ns, not significant; SDS-PAGE, sodium dodecyl sulfate-polyacrylamide gel electrophoresis; RBD, receptor-binding domain; KD, equilibrium dissociation constant; SD, standard deviation; Co-IP, coimmunoprecipitation; chECL1, chicken first extracellular loop of chNHE1; chNHE1, chicken sodium hydrogen exchanger isoform 1.

loop, effectively anchoring the two ends of the loop on its both sides (Fig 4A). However, when Asn123 of gp85 was substituted with Ile123, the formation of non-polar amino acids and hydrogen bonds with neighboring amino acids ceased, increasing the rigidity of the structure (Fig 4B). The structure of TM-chECL1 was shown in Fig 4C. Analysis of the interaction at the interface between HPRS103-gp85 and TM-chECL1, which comprised seven amino acids (Trp145, Lys113, Lys144, Tyr112, His122, Ser147, and Ile152) of HPRS103-gp85 and five amino acids (Gln40, Trp38, Thr37, Gly43, and Trp42) of TM-chECL1, showed that Lys113 and Gln40, Tyr112 and Trp38, His122 and Thr37, and Ser147 and Gly43 formed four hydrogen bonds (Fig 4D). However, the binding interface between N123I-gp85 and TM-chECL1 was not identical to that between HPRS103-gp85 and TM-chECL1. The binding interface of N123I-gp85 with TM-chECL1 primarily comprised eight amino acids (Thr101, Arg102, Cys104, Thr106, Ser170, Val120, His122, and Ile123) of N123I-gp85 and ten amino acids (Val33, Arg32, Ser34, Trp38, Glu35, Gln40, Trp42, Thr37, Glu39, and Thr31) from TM-chECL1, and formed six hydrogen bonds (Arg102 with Val33 and Ser34, Ser170 with Glu35, Ile123 with Gln40, Thr106 with Thr37, and Thr101 with Thr31) (Fig 4E). Molecular dynamics simulations were performed using the Poisson-Boltzmann surface area method for molecular mechanics to calculate the binding free energies of HPRS103-gp85 or N123I-gp85 complexed with TM-chECL1. Each simulation was performed at 100 ns and each model was simulated in triplicate. All trajectories reached a plateau in root-mean-square deviation after 50 ns (Fig 5A and 5B), indicating that their structures reached equilibrium. The ΔG of the HPRS103-gp85 was significantly high (-300 kJ/mol) (Fig 5C), approximately 2-fold higher than that for N123I-gp85 (-600 kJ/mol) (Fig 5D), which suggested that N123I mutation was closely related to the improved binding capacity of gp85 to chNHE1.

## The N123I mutation of gp85 enhances the replication ability of ALV-J

To further validate the impact of the N123I mutant of gp85 on the binding capacity and replication ability of ALV-J, an ALV-J variant (rHPRS103-N123I) containing only the N123I mutation on the gp85 glycoprotein was constructed and rescued using rHPRS103 as the backbone (Fig 6A). Western blotting results showed that rHPRS103-N123I produced a specific 27-kDa band with the 2E5 monoclonal antibody (Fig 6B). The IFA results also indicated that DF1 cells inoculated with rHPRS103-N123I showed specific green fluorescence, but uninfected cells did not show such fluorescence (Fig 6C).

To determine the impact of the N123I mutation of gp85 on the binding capacity of HPRS103, both 50 MOI rHPRS103-N123I and rHPRS103 were separately inoculated with 293T cells expressing transient chNHE1. The virus-cell binding capacity was assessed through RT-qPCR and FACS experiments, revealing that rHPRS103-N123I had binding capacity to chNHE1 7.14-fold and 1.27-fold greater than that of rHPRS103 to chNHE1, respectively (Fig 6D and 6E). To further investigate whether the N123I mutation in gp85 enhances the

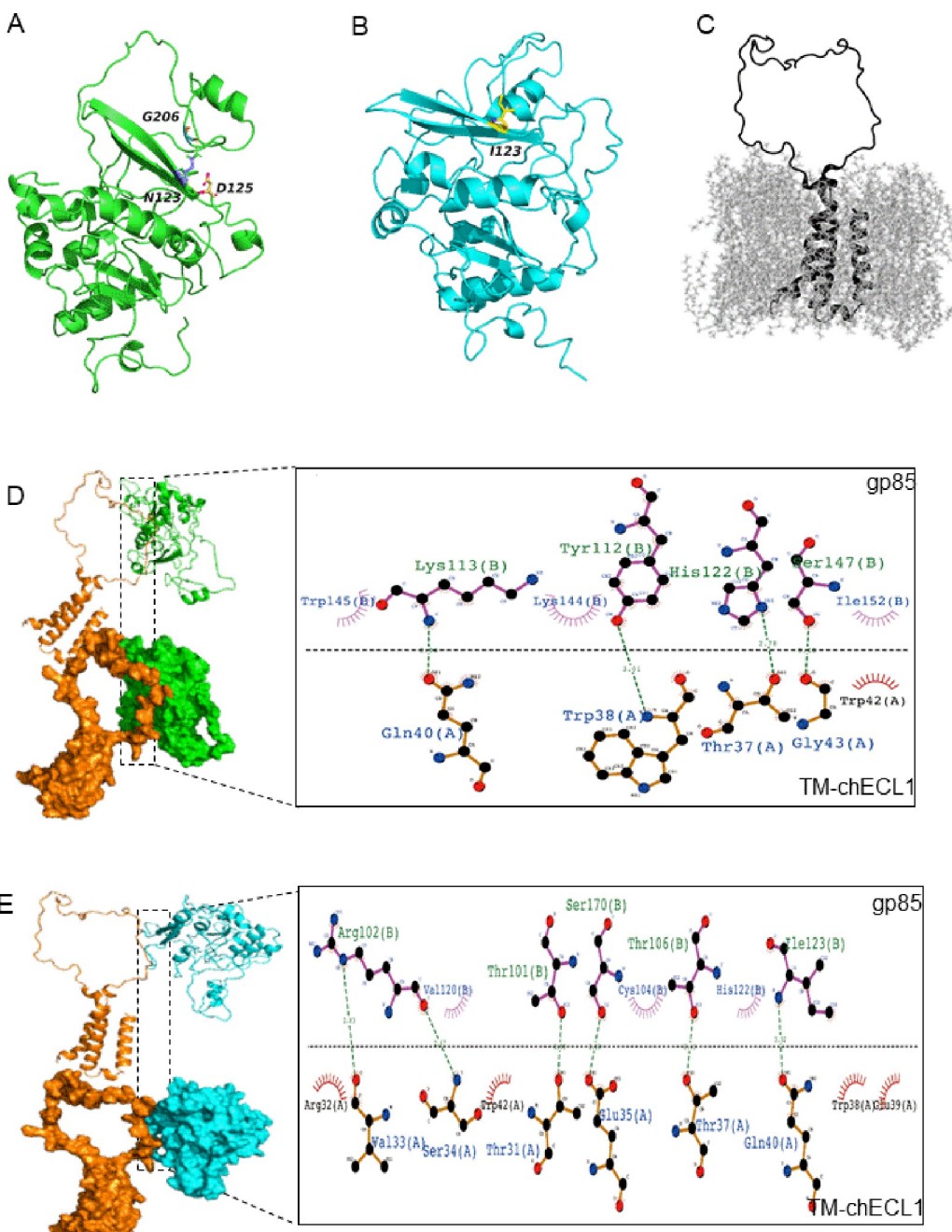

**Fig 4. The modeled structure of N123I-gp85, HPRS103-gp85, and TM-chECL1 proteins, and comparison of molecular docking of both gp85 proteins with TM-chECL1.** (A) The HPRS103-gp85 structure is presented as a ribbon. The side chains of Asn123, Asp125, and Gly206 are displayed. (B) The N123I-gp85 structure is presented as a ribbon. The side chains of Ile123 are displayed. (C) The TM-chECL1 structure is presented as a ribbon. (D) The interaction between HPRS103-gp85 and TM-chECL1 is analyzed using pymol and Ligplot+. (E) The interaction of N123I-gp85 with TM-chECL1 is analyzed by pymol and Ligplot+. The chain on the upside represents gp85; the chain on the downside represents TM-chECL1; the green line represents the hydrogen bond. TM-chECL1, chECL1 with transmembrane region proteins.

replication of HPRS103, DF1 cells were inoculated separately with 0.01 MOI of rHPRS103-N123I and rHPRS103. Cells were harvested at 1, 2, 3, 4, 5, 6, and 7 dpi. The growth curve showed a 1.5- to 8.2-fold increase in viral titers of rHPRS103-N123I at 3 to 7 dpi

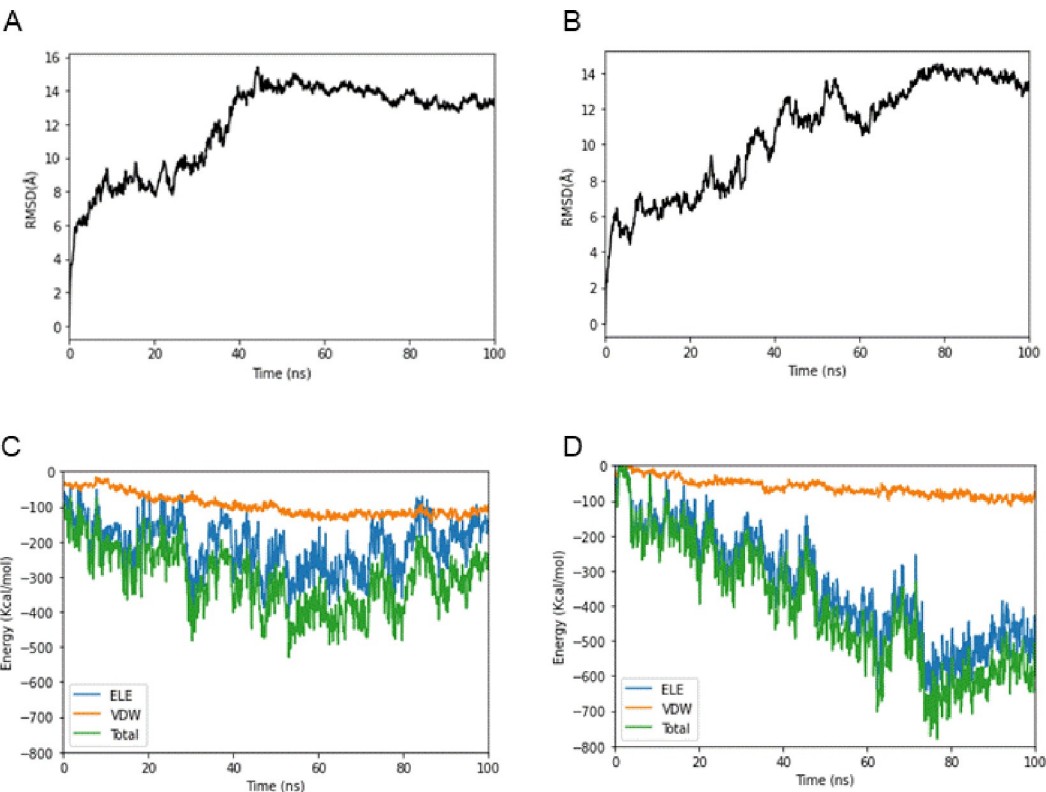

**Fig 5. Binding free energy calculated for the N123I-gp85, HPRS103-gp85, and TM-chECL1 proteins.** (A) RMSDs of the backbone atoms of HPRS103-gp85-TM-chECL1 complexes. (B) RMSDs of the backbone atoms of N123I-gp85-TM-chECL1 complexes. (C) Binding free energy of HPRS103-gp85 with TM-chECL1. (D) Binding free energy of N123I-gp85 with TM-chECL1. RMSD, root-mean-square deviation; TM-chECL1, chECL1 with transmembrane region proteins.

compared to that of rHPRS103 (Fig 6F). The RT assay demonstrated that the RT activity of rHPRS103-N123I was significantly greater than rHPRS103 during the 3–7 dpi period, as illustrated in Fig 6G, and consistent with viral titer results. Furthermore, we observed *in vivo* replication of both rHPRS103 and rHPR103-N123I viruses following intraperitoneal injection of a $10^4$ TCID$_{50}$ dose into 1-day-old specific-pathogen-free chickens. The results of the viremia indicated that the viral loads of the rHPRS103-N123I groups were 2.2 to 13.4 times higher than the rHPRS103 group (Fig 6H). Moreover, cloacal viral shedding signified that SPF chickens who were infected with rHPRS103-N123I exhibited viral shedding at 14 dpi, earlier than SPF chickens infected with rHPRS103, which showed viral shedding at 17 dpi. In addition, the proportion of cloacal swabs testing positive for ALV in the rHPRS103-N123I group remained consistently high (80%) from 18 days post-infection and throughout the duration of the study. This percentage was higher than that observed in the rHPRS103 group (66.7%) (Fig 6I), indicating that the gp85 N123I mutant increases the replication capacity of ALV-J both *in vitro* and *in vivo*.

## Discussion

ALV-J gp85 is prone to mutation to gain an evolutionary advantage against natural selection [36]. Previous research has highlighted significant variations in the gp85 sequences of ALV-J layer strains, particularly in the RBD region, compared to the ALV-J broiler strain [18,30]. In this study, we systematically analyzed the relationship between amino acid mutations in gp85 and the ability of the virus to replicate. First, the replication ability of the layer strain

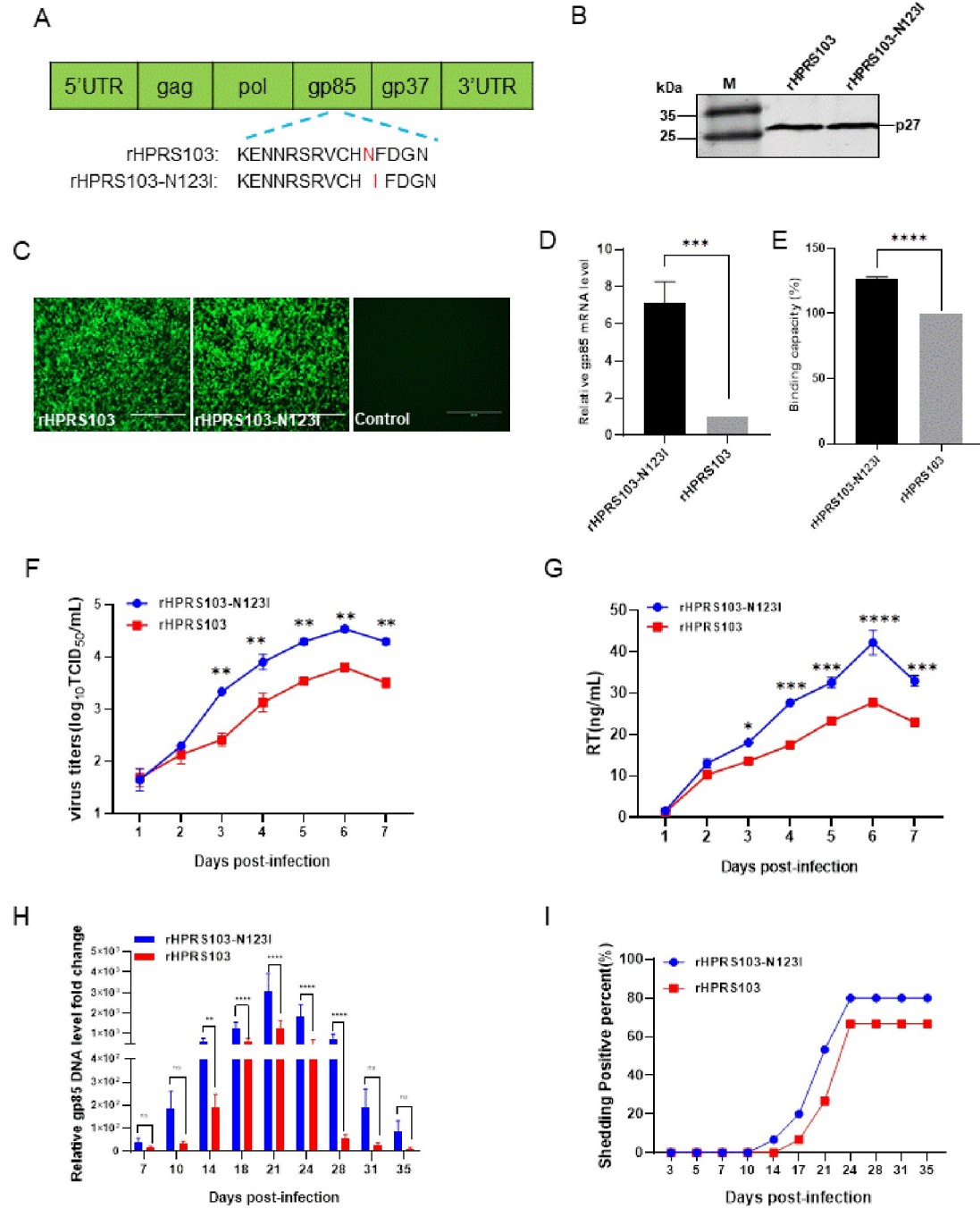

**Fig 6. The replication ability of the recombinant viruses rHPRS103-N123I and rHPRS103 *in vivo* and *in vitro*.** (A) The schematic diagram shows the construction of the recombinant virus, rHPRS103-N123I. Using rHPRS103 as the backbone, the Asn123 of rHPRS103 is mutated to Ile123 for constructing the recombinant virus rHPRS103-N123I. (B) Western blotting. The recombinant viruses rHPRS103-N123I and rHPRS103 are detected with the 2E5 monoclonal antibody, rHPRS103 as reference. (C) IFA assay. DF-1 cells are infected with recombinant viruses, rHPRS103-N123I and rHPRS103, for 72 h, and then detected with the 4A3 monoclonal antibody and analyzed using a fluorescence microscope, rHPRS103 as reference. Scale bar: 400 μm. (D and E) Binding capacity between the rHPRS103-N123I and rHPRS103 with chNHE1 expressed on the cell membrane of transfected 293T cells is detected by RT-qPCR (D) and FCAS assays (E). (F) The replication ability of rHPRS103-N123I and rHPRS103 is detected using $TCID_{50}$ *in vitro*. DF1 cells are incubated with rHPRS103-N123I and rHPRS103 at an MOI of 0.01, and then harvested and quantified at 1, 2, 3, 4, 5, 6, and 7 dpi. (G) The replication ability of rHPRS103-N123I and rHPRS103 is detected using an RT assay. DF1 cells are incubated with rHPRS103-N123I and rHPRS103 at an MOI of 0.01, and then harvested and quantified 1, 2, 3, 4, 5, 6, and 7 dpi. (H and I) One-day-old SPF chickens were injected intraabdominally with rHPRS103-N123I and rHPRS103 (n = 15, dose = $10^4$ $TCID_{50}$). (H) Viral loads in whole-blood samples are collected and

evaluated using real-time SYBR qPCR mix PCR at different time points. (I) Cloaca swabs are collected and evaluated by ELISA at different time points. The error bars represent the SD. *, $P < 0.05$; **, $P < 0.01$; ***, $P < 0.001$; ns, not significant; ELISA, enzyme-linked immunosorbent assay; IFA, immunofluorescence; TCID$_{50}$, 50% tissue culture infective dose; dpi, days post-inoculation; MOI, multiplicity of infection; SPF, specific pathogen-free; qPCR, quantitative polymerase chain reaction; SD, standard deviation.

JL08CH3-1 is superior to that of the prototype strain HPRS103. Furthermore, JL08CH3-1-gp85 had a stronger capacity to bind to the receptor chNHE1 than HPRS103-gp85. Crucially, the Asn123 mutation to Ile123 in the RBD region of gp85 increased its stability, reduced the binding energy required for interaction with chNHE1, and increased the number of amino acids and hydrogen bonds at the interaction interface, improving the binding capacity between gp85 and chNHE1, and increasing viral replication ability both *in vivo* and *in vitro*.

The binding of the retrovirus envelope protein to its specific receptor on target cells is the most crucial stage of infection [37]. The specific interaction between ALV-J gp85 and chNHE1 triggers conformational changes in the structure of the trimeric envelope glycoprotein, leading to the exposure of the fusion peptide in gp37, allowing it to interact with the target cell membrane and mediate viral invasion [27,38–40]. During natural selection, different subgroups of ALV enhance their ability to infect host cells by altering receptor usage through mutations or recombination of the envelope protein [25,41]. Although the receptor usage of the ALV-J layer strains remains unchanged [23,42,43], significant genetic variation in their gp85 sequences, with less than 90.6% homology with the prototype strain HPRS103 [18,30]. Our study revealed a close correlation between amino acid variations in gp85 and the replication ability of ALV-J. Other studies have shown that the RBD region plays a critical role in the binding of viral envelope proteins to the receptor, directly affecting viral replication. For example, the A82V mutation in the RBD region of the GP protein of the Ebola virus (EBOV) improves its ability to bind to human NPC1, enhancing the infectivity of EBOV in humans and causing extensive outbreaks, such as those seen in Liberia, Sierra Leone, and Guinea [44,45]. The G100R mutation in the RBD region of the murine leukemia virus increases the binding capacity between the SU protein and the Pit-2 receptor, accelerating the fusion efficiency between the virus and cells, and promoting virus replication in host cells [46]. Ser375 is replaced by a hydrophobic or basic amino acid in the Env of the human immunodeficiency virus or Simian immunodeficiency virus, which increases the binding capacity of Env to the CD4 receptor and improves viral replication ability in the host [33]. The E484K and D614G mutations in the SARS-CoV-2 RBD increase its capacity to bind to the ACE2 receptor and enhance its replication in human lung cell lines (Calu-3) and primary human respiratory tissues [47–50]. Our previous study identified the RBD region of ALV-J gp85, which is located on the bipartite sequence motif, including amino acids 38–131 in the N-terminus (including Vr1, Vr2, and hr1) and amino acids 159–283 in the C-terminus (including hr2 and V3) of gp85 [35]. Our results demonstrated that N123I in the hr1 region of the N-terminus of the RBD is a key amino acid that increases the binding capacity between the gp85 protein and chNHE1, and enhances the replication ability of ALV-J. These findings further indicate that hr1 at the N-terminus of the RBD region of ALV is a critical area that influences the binding capacity between gp85 and the receptor.

The receptor-binding ability of the viral envelope protein RBD depends on the stability, hydrogen bonding, and electrostatic potential of the envelope protein. Mutations such as D614G and E484K in SARS-CoV-2 and G100R in murine leukemia virus have been shown to affect RBD binding capacity. [46,47,51]. To uncover the potential mechanism by which the N123I mutation improves the binding capacity between gp85 and chNHE1, we developed a structural model and conducted molecular docking analysis. Our findings suggest that mutating Asn123 to Ile123 may enhance the stability of the N-terminal RBD region of the gp85

protein and reduce steric clashes, resulting in a greater likelihood of binding between gp85 and the chNHE1 receptor. Furthermore, Ile123 was present at the interaction interface and established a hydrogen bond with Gln40 of TM-chECL1. This expanded the binding interface of gp85 with TM-chECL1 and resulted in a decrease in the binding free energies (by 300 kJ/mol). This phenomenon has also been observed in the binding of ALV-A [52] and SARS-CoV-2 [53] to their respective receptors. Thus, it is plausible that the N123I mutation could increase the binding capacity of gp85 to the chNHE1 receptor by strengthening the structure of the stability of the gp85 protein, expanding the docking surface, and reducing binding free energies.

As an important vertically-transmitted virus, ALV-J has been spreading and circulating worldwide since its isolation in 1988 [54]. ALV-J is the dominant ALV subgroup in chickens, which seriously affects the safety of poultry breeding sources in China [19]. In this study, the N123I mutation improved the replication of ALV-J. To determine whether the N123I mutation in the gp85 RBD region is closely related to the transmission of ALV-J, we systematically analyzed the gp85 sequences of ALV-J isolated from different countries from 1990 (Fig 7). The N123I mutation was first found in broiler strains in the United States in 2000, coinciding with the widespread outbreak of ALV-J in the country [55]. Second, the N123I mutation emerged in ALV-J layer strains in 2008 in China, coinciding with the extensive ALV-J epidemic among layer chickens in 2008–2009, causing significant economic losses to the poultry industry at that time [17,30,56–59]. In recent years, as ALV has been eradicated from layers and broiler chicken flocks, the ALV-J infection rate in these flocks was relatively low in China. However, the ALV-J infection rate in local chickens is relatively high [20,60,61]. The N123I mutation was also detected in local chicken isolates and its detection rate gradually increased. We found that the occurrence of the N123I mutation coincided with the timeline of the ALV-J epidemic outbreak. Furthermore, the animal experimental results showed that the Asn123 mutation to Ile123 within the gp85 RBD of HPRS103 improved its replication ability, resulting in earlier shedding and a higher shedding rate. Therefore, we speculate that the N123I mutation enhanced the replication ability of ALV-J, making it more easily transmitted and likely to cause large epidemics. This finding is like that of the A82V mutation in the GP of EBOV, which led to the epidemic of 2013–2016 [62,63]. Based on the results and the above analysis, we suggest that in future epidemiological investigations, more attention should be paid to amino acid mutations in the RBD region of ALV-J to provide an early warning of another large-scale outbreak of ALV-J.

In summary, our study demonstrated that the stronger binding capacity between gp85 of the layer strain JL08CH3-1 and chNHE1 was a key factor in enhancing the replication ability of JL08CH3-1 compared to that of the prototype strain HPRS103. We revealed for the first time that the N123I mutation within the RBD region of gp85 plays a crucial role in enhancing the binding capacity between gp85 and the chNHE1 receptor, as well as enhancing the replicative ability of ALV-J, both *in vitro* and *in vivo*. Epidemiological investigations and experimental animal data have confirmed that the N123I mutation enhances the replication advantage of ALV-J, making it more prone to infection and transmission within the host. Our study clarified the molecular mechanism by which amino acid mutations within the RBD region of gp85 enhance its replication ability and emphasized the importance of continuously monitoring variations within the RBD region of ALV-J gp85.

## Materials and methods

### Ethics statement

All animal experiments were approved by the Committee on the Ethics of Animal Experiments of Harbin Veterinary Research Institute (HVRI), Chinese Academy of Agricultural Sciences

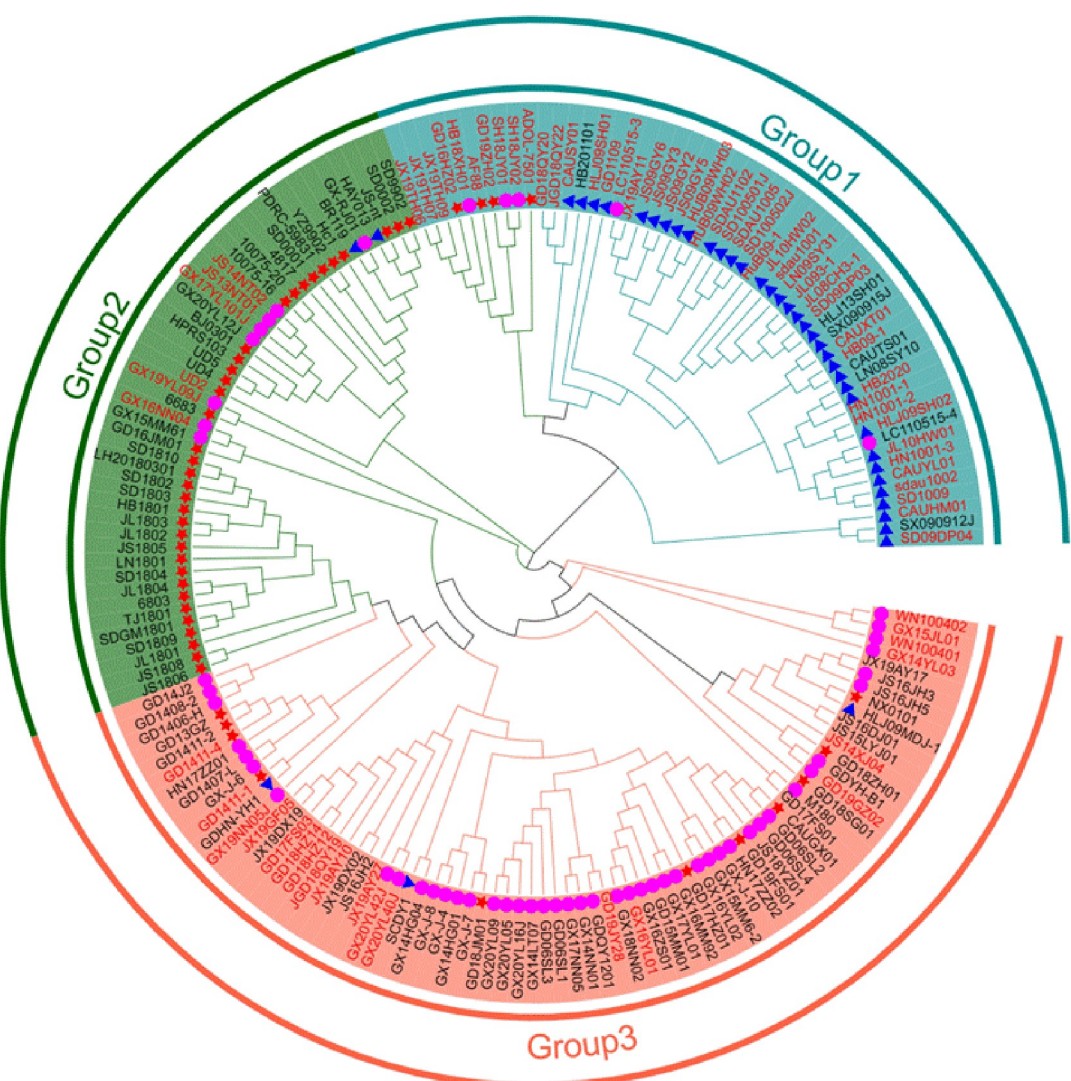

**Fig 7. Phylogenetic tree analysis of the evolutionary relationships between the *gp85* gene in various ALV-J strains.** The tree is constructed using the neighbor-joining method with MEGA 6.0 software (https://www.megasoftware.net/home). Bootstrap values are calculated with 1,000 repetitions of the alignment. The three groups are marked. Pink-purple circles represent ALV-J local strains. The blue triangles represent ALV-J layer strains. The red stars represent the ALV-J broiler strains. The N123I mutations within the RBD region of the virus are marked in red. All sequences of ALV-J strains are obtained from GenBank. The accession numbers are listed in the S1 Table.

(CAAS). Specific pathogen-free (SPF) chickens were purchased from the Experimental Animal Centre of the HVRI and housed in negative-pressure isolators with adequate food and light. All animal procedures were performed according to the international standards for animal welfare.

## Cells and viruses

293T cells and chicken embryonic fibroblast DF-1 cells were maintained in Dulbecco's modified Eagle's medium (DMEM) (L110KJ, BasalMedia, Shanghai, China) supplemented with 10% (vol/vol) heat-inactivated fetal bovine serum (FBS) (FDN500, Excell Bio, Shanghai, China) and 1% (vol/vol) penicillin/streptomycin (15140122, Gibco, Waltham, MA, USA) in a

humidified incubator with 5% $CO_2$ at 37˚C. The ALV-J prototype strain HPRS103 was kindly provided by Prof. Venugopal Nair and the layer strain JL08CH3-1 was maintained at the Harbin Veterinary Research Institute, CAAS (Harbin, China).

## Plasmids

Eukaryotic expression plasmids (CAG-sgp85-Fc and pCAF-gp85) for soluble HPRS103-gp85, chNHE1 (pCAF-chNHE1), and chECL1 (with an HA tag for C-terminal fusion) were constructed as described in our previous study [35]. To compare the binding capacity between HPRS103-gp85 and JL08CH3-1-gp85 and the chNHE1 receptor, we constructed eukaryotic expression plasmids of JL08CH3-1-gp85 using a previously-established strategy. Briefly, the JL08CH3-1-gp85 gene was amplified from the proviral DNA of the layer strain JL08CH3-1 and cloned into the CAG plasmid (Addgene, Cambridge, MA, USA) or the pCAGGS plasmid with a constant region fragment of human IgG or a FLAG tag at the N-terminus to generate CAG-JL3-1-gp85-Fc or pCAF-JL3-1-gp85 plasmids, respectively.

To construct the gp85 mutant plasmids, we divided the 23 nonsynonymous mutations within the RBD of HPRS103-gp85 and JL08CH3-1-gp85 into 13 groups, following the substitution method reported in a previous study [35]. Subsequently, using the CAG-sgp85-Fc plasmid as the backbone, the amino acids within each group were replaced with the corresponding amino acids from JL08CH3-1-gp85. To further investigate which amino acid mutation in N123I/V128F plays a significant role in affecting the binding capacity of the gp85 protein to chNHE1, we constructed N123I-gp85 (CAG-N123I-gp85-Fc and pCAF-N123I-gp85) and V128F-gp85 (CAG-V128F-gp85-Fc) single mutant plasmids.

To construct pBlue-HPRS/JL3-1 infectious clones, we used pBlue-HPRS103 infectious clones as the backbone and replaced gp85 with gp85 from JL08CH3-1. Similarly, to construct pBlue-HPRS103-N123I infectious clones, we used pBlue-HPRS103 infectious clones as the backbone and replaced the amino acid Asn123 with Ile123.

All the constructs were verified by DNA sequencing. The primer sequences for all oligonucleotides used in this study are available upon request.

## Purification and quantification of gp85 mutants and soluble chECL1

293T cells were transfected with CAG-sgp85-Fc, CAG-JL3-1-gp85-Fc, mutant gp85, or soluble chECL1 plasmids using a PEI transfection reagent (23966, Polysciences, Shenzhen, China) according to the manufacturer's instructions. At 48 h post-transfection, the cell culture medium was harvested and centrifuged at 4˚C ($8,000 \times g$ for 5 min) to remove cell debris. The supernatant was purified using Protein A Resin (L00210, Gen Script, Piscataway, NJ, USA) according to the manufacturer's protocol. Different proteins were labeled and purified separately to avoid cross-contamination. Expression levels and molecular weights of purified proteins were analyzed using 12.5% sodium dodecyl sulfate-polyacrylamide gel electrophoresis (SDS-PAGE) with Coomassie blue staining. The concentration of purified proteins was quantified using a bicinchoninic acid protein assay kit (23227, Thermo Scientific, Waltham, MA, USA).

## Protein-cell binding assay

Protein-cell binding assay was performed as previously described [23]. Briefly, 293T cells were transfected with the pCAF-chNHE1 plasmid. After 24 h post-transfection (hpi), 293T cells expressing chNHE1 were digested with EDTA-trypsin, centrifuged for 10 min ($1,000 \times g$) at 4˚C, and then washed three times with ice-cold PBS containing 5% (w/v) FBS. In parallel, 0.5 mL of 200 ng/μL wild-type and mutant gp85 proteins were incubated with cells for 1 h on ice,

followed by washing three times with ice-cold PBS containing 5% (w/v) FBS. Cells were then stained with a 1:200 dilution of goat anti-human IgG (Fc specific)-FITC antibody (F9512, Sigma-Aldrich, St. Louis, MO, USA) for 1 h on ice. After washing three times with ice-cold PBS containing 5% (w/v) FBS, the cells were fixed in 4% (v/v) paraformaldehyde (PFA) (P1110, Solarbio, Beijing, China) at 25˚C for 15 min and then washed three times. Cells were stained with a 1:200 dilution of anti-FLAG M2 monoclonal antibody produced in mice (F1804, Sigma-Aldrich, St. Louis, MO, USA) for 1 h on ice and then washed three times. The cells were then incubated with a 1:200 dilution of anti-mouse IgG-TRITC antibodies (T2402, Sigma-Aldrich, St. Louis, MO, USA) for 1 h on ice. After washing three times, the cells were resuspended in 0.5 mL ice-cold PBS containing 5% (w/v) FBS and analyzed by FACS using a FACS ARIA II flow cytometer (Cytomics FC 500, BD Biosciences, San José, CA, USA).

## Co-IP experiments

293T cells were seeded in 6-well plates and transfected with the 2 μg respective plasmids by using the TransIT-X2 Dynamic Delivery system (MIR6000, Mirus Bio LLC, USA) according to the manufacturer's instructions. At 48 hpi, the cells were washed three times with ice-cold PBS and then lysed in 0.2 mL western blotting and IP lysis buffer (P1003, Beyotime, Haimen, China) for 30 min on ice. After $12,000 \times g$ centrifugation for 10 min at 4˚C, the supernatant was incubated with anti-HA-agarose Mab (A2095, Sigma-Aldrich, St. Louis, MO, USA) at 4˚C overnight or 6–8 h. The beads were collected $4,500 \times g$ centrifugation for 5 min at 4˚C and washed five times with ice-cold PBS. Immunoprecipitated proteins were separated using 12.5% SDS-PAGE and detected by Western blotting.

## Western blotting

Protein and cell samples were boiled in $5 \times$ SDS loading buffer (P10015L, Beyotime, Shanghai, China) at 100˚C for 10 min, separated by 12.5% SDS-PAGE, and blotted onto nitrocellulose membranes (66485, PALL, Port Washington, NY, USA). The membrane was blocked for 1.5 h with 5% (w/v) skim milk (232100, BD Pharmingen San Diego, CA, USA) in PBS at room temperature. After washing three times with PBST (PBS containing 0.1% Tween 20), the membranes were exposed to primary antibodies in PBST for 1.5 h. The membranes were then washed three times using PBST. After 1 h of exposure to IRDye800CW goat anti-mouse IgG (H+L) antibody (926–32210, LI-COR, Lincoln, NE, USA), the membranes were washed and analyzed using the LI-COR Odyssey Imaging System (Li-Cor Biosciences). All experiments were performed at least in triplicate.

## Rescue of recombinant viruses

The highly purified plasmids pBlue-HPRS/JL3-1, pBlue-HPRS103 and pBlue-HPRS103-N123I were introduced respectively into DF-1 cells using the TransIT-X2 Dynamic Delivery system (MIR6000, Mirus Bio LLC, USA). The culture supernatant containing the virus stocks was harvested 7 days later and then blind-passaged into secondary DF-1 cells. The rescued viruses were named rHPRS/JL3-1, rHPRS103, and rHPRS103-N123I. The titer of the rescued recombinant virus was determined based on the $TCID_{50}$ value using the Reed and Muench method [64].

## Virus-cell binding assay

The virus-cell binding experiment followed established procedures with slight modifications [65]. In summary, 293T cells were transfected with the pCAF-chNHE1 plasmid and left to incubate for 24 h. Then they were cooled to 4˚C for 10 min and exposed to rescued viruses at

an MOI of 50, including rHPRS/JL3-1, rHPRS103, and rHPRS103-N123I, for 1 h at 4˚C. The cells were then washed five times on ice and collected for quantification of the binding capacity of the ALV-J virus by RT-qPCR. Total RNA from the cell samples using RNAiso Plus (9109, Takara Bio Inc, Dalian, China) and reverse transcribed into cDNA using reverse transcriptase (R223-01, Vazyme, Nanjing, China). A fragment of the gp85 gene of ALV-J was used as the real-time PCR (qPCR) target gene [66] and the human glyceraldehyde-3-phosphate dehydrogenase gene was used as an internal reference gene in the host cell genome [67]. qPCR was performed using a fluorescent qPCR instrument (Quant Studio 5, Applied Biosystems Waltham, MA, USA) under the following cycling conditions: 95˚C for 1 min for initial denaturation, followed by 40 cycles of 95˚C for 15 s for denaturation, 60˚C for 1 min, and collection of PCR product signals. The results were analyzed using the $2^{-\Delta\Delta Ct}$ method.

FACS was also used to assess the binding capacity of ALV-J virus-cells. To achieve this, 293T cells were transfected with the pCAF-chNHE1 plasmid. After 24 hpi, the 293T cells were treated with EDTA-trypsin, centrifuged for 10 min (1,000 × $g$) at 4˚C, and washed thrice using ice-cold PBS containing 5% (w/v) FBS. Then 50 MOI rescued viruses such as rHPRS/JL3-1, rHPRS103, or rHPRS103-N123I were incubated with cells on ice for 1 h thereafter, cells were washed thrice using ice-cold PBS that contained 5% (w/v) FBS. Cells were stained with a 1:200 dilution of 4A3, a mouse anti-gp85 antibody, in ice-cold PBS for 1 hour. After three washes with ice-cold PBS supplemented with 5% (w/v) FBS, the cells underwent incubation with a 1:200 concentration of FITC-conjugated goat anti-mouse IgG secondary antibody (F0257, Sigma-Aldrich, St. Louis, MO, USA) also at 1 h on ice. Subsequently, cells were washed three times with ice-cold PBS containing 5% (w/v) FBS. Cells were fixed in 4% paraformaldehyde (PFA) (P1110, Solarbio, Beijing, China) at 25˚C for 15 min. The cells were then washed three times. Subsequently, cells were washed three times, followed by staining with a 1:200 dilution of anti-FLAG M2 monoclonal antibody produced in Rabbit (SAB4301135, Sigma-Aldrich, St. Louis, MO, USA) for 1 h on ice. The cells were incubated with a 1:200 dilution of Alexa Fluor 546 goat anti-Rabbit IgG (H+L) antibodies (A11010, Thermo Scientific, Waltham, MA, USA) for 1 h on ice. After washing three times, they were resuspended in 0.5 mL ice-cold PBS containing 5% (w/v) FBS and analyzed by FACS with a FACS ARIA II flow cytometer (Cytomics FC 500, BD Biosciences, San José, CA, USA).

## Virus growth curves and reverse transcriptase (RT) activity

DF1 cells were infected with ALV-J at an MOI of 0.01. Infected cells and supernatants were harvested at 1, 2, 3, 4, 5, 6, and 7 dpi. The Reed and Muench method was used to calculate viral titers [64]. The harvested samples were also quantitated the RT activity by using a colorimetric reverse transcriptase assay (11468120910, Roche Applied Science, Indianapolis, USA) according to the manufacturer's protocol.

## Indirect immunofluorescent assay (IFA)

The IFA assay was performed as previously described [68]. First, DF-1 cells were respectively infected with 200 $TCID_{50}$ rHPRS/JL3-1, rHPRS103, and rHPRS103-N123I at 37˚C with 5% $CO_2$ for 2 h. After incubation, cells were washed three times with PBS at room temperature and kept in DMEM containing 5% (w/v) FBS at 37˚C and 5% $CO_2$ for 72 h. Next, the cells were washed with cold PBST three times and fixed with cold solute ethanol for 15 min at room temperature. After washing three times, the cells were incubated with 4A3 (mouse anti-gp85 antibody) at a dilution of 1:200 in PBS for 1 h at 37˚C. This was followed by washing and then incubation with 1:200 dilution of a fluorescein isothiocyanate-conjugated secondary antibody (FITC-conjugated goat anti-mouse IgG antibody) for 1 h at 37˚C. After three washes with

PBST, DF-1 cells were visualized using a fluorescence microscope (TU-80, Nikon, Tokyo, Japan). Normal DF-1 cells were used as negative controls.

## Surface plasmon resonance analysis

chECL1 protein was immobilized on a CM5 chip (BR100530, GE Healthcare, Waukesha, WI, USA) to a level of approximately 100 response units using Biacore 8K (Cytiva, Marlborough, MA, USA) and a running buffer (10 mM HEPES pH 7.2, 150 mM NaCl, and 0.05% Tween-20). Serial dilutions of HPRS103-gp85, JL08CH3-1-gp85, and N123I-gp85 were passed through a CM5 chip. The resulting data were fitted to a 1:1 binding model using Biacore Evaluation Software (Cytiva, Marlborough, MA, USA).

## Structural analysis

Protein structures of HPRS103-gp85, N123I-gp85, and TM-chECL1 were predicted using the open-source server trRosseta (https://yanglab.nankai.edu.cn/trRosetta/) [69]. The model was optimized by using molecular dynamics simulation. All simulations and analyses were performed using GROMACS (https://www.gromacs.org). The initial docking models for TM-chECL1, HPRS103-gp85, and N123I-gp85 were constructed using the Hdock server (http://hdock.phys.hust.edu.cn/) [70]. The highest-scoring model was further optimized using Rosetta 2020 (https://yanglab.nankai.edu.cn/trRosetta/) to obtain the final docking model. The protein-protein interacting faces were analyzed using pmyol and Ligplot+ [71]. The free energies of TM-chECL1 and HPRS103-gp85 or N123I-gp85 binding were obtained by molecular dynamics simulation using GROMACS based on the final docking model.

## Experimental infections

The replication abilities of rHPRS103 and rHPRS103-N123I were evaluated in SPF chickens. A total of 45 one-day-old SPF chickens were randomly divided into three groups of 15 chickens each and maintained separately in a negative-pressure isolator. The two groups received an intraperitoneal injection of $10^4$ $TCID_{50}$ per chicken of rHPRS103 or rHPRS103-N123I, respectively. One group of chickens formed a control group and were inoculated with DMEM. Cloacal swabs and whole-blood samples were collected from all chickens in each of the three groups at 3, 5, 7, 10, 14, 17, 21, 24, 28, 31, and 35 dpi. Cloacal swabs were examined using an Avian Leukosis Virus Antigen Test (ELISA) kit (NEE83500, Sinaean Biologics, Harbin, China) according to the manufacturer's instructions. Whole-blood samples were analyzed using qPCR.

## qPCR

To detect viral loads in whole-blood samples, DNA was extracted using a Body Fluid Viral DNA/RNA Miniprep Kit (AP-MN-BF-VNA-250, Axygen Union City, CA, USA). Viral loads in whole-blood samples were detected by real-time PCR using the THUNDERBIRD SYBR qPCR Mix Kit (QPS-201, TOYOBO, Osaka, Japan). A fragment of the gp85 gene of ALV-J was used as the real-time PCR (qPCR) target gene [66] and the chicken ovotransferrin gene was used as an internal reference gene in the host cell genome [72]. qPCR was performed using a fluorescent qPCR instrument (Quant Studio 5, Applied Biosystems Waltham, MA, USA) under the following cycling conditions: 95˚C for 1 min for initial denaturation, followed by 40 cycles of 95˚C for 15 s for denaturation, 60˚C for 1 min, and collection of PCR product signals. The results were analyzed using the $2^{-\Delta\Delta Ct}$ method.

## Statistical analysis

GraphPad Prism software (version 7.03, GraphPad Software, San Diego, CA, USA) was used for the statistical analysis. The student's $t$-test was used to assess differences between groups. Statistical significance was established at $P < 0.05$.

## Supporting information

**S1 Table. The ALV-J strains were used in the study.**
(PDF)

## Acknowledgments

We thank Prof. Venugopal Nair (the Pirbright Institute) for the ALV-J prototype strain HPRS103.

## Author Contributions

**Conceptualization:** Yulong Gao.

**Data curation:** Mengmeng Yu, Yao Zhang, Yulong Gao.

**Formal analysis:** Mengmeng Yu, Yao Zhang, Yulong Gao.

**Funding acquisition:** Suyan Wang, Yulong Gao.

**Investigation:** Mengmeng Yu, Yao Zhang, Li Zhang, Suyan Wang, Zhuangzhuang Xu, Yulong Gao.

**Methodology:** Mengmeng Yu, Yao Zhang, Li Zhang, Suyan Wang, Yongzhen Liu, Zhuangzhuang Xu, Peng Liu, Yuntong Chen, Ru Guo, Xiaole Qi, Li Gao, Yanping Zhang, Hongyu Cui, Yulong Gao.

**Project administration:** Yulong Gao.

**Resources:** Mengmeng Yu, Yao Zhang, Li Zhang, Yongzhen Liu, Zhuangzhuang Xu, Peng Liu, Lingzhai Meng, Tao Zhang, Wenrui Fan.

**Software:** Mengmeng Yu, Yao Zhang, Li Zhang, Ru Guo.

**Supervision:** Yulong Gao.

**Validation:** Peng Liu, Yuntong Chen.

**Visualization:** Mengmeng Yu, Yao Zhang, Yulong Gao.

**Writing – original draft:** Mengmeng Yu, Yulong Gao.

**Writing – review & editing:** Suyan Wang, Yongzhen Liu, Yulong Gao.

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
