## [Decision Letter · Decision Letter 0]

30 Sep 2023

Dear Prof. Gao,

Thank you very much for submitting your manuscript "N123I mutation in the ALV-J receptor-binding domain region enhances the viral replication ability by increasing binding affinity with chNHE1" for consideration at PLOS Pathogens. As with all papers reviewed by the journal, your manuscript was reviewed by members of the editorial board and by several independent reviewers. In light of the reviews (below this email), we would like to invite the resubmission of a significantly-revised version that takes into account the reviewers' comments.

The reviewers agreed that your findings are of interest and potential significance in the evolution of the ALV-J strain. However, reviewer 1 raises significant concerns on certain fundamental aspects of the experimental design, in particular the MOI used: these points need to be addressed before a revised manuscript can be considered.

We cannot make any decision about publication until we have seen the revised manuscript and your response to the reviewers' comments. Your revised manuscript is also likely to be sent to reviewers for further evaluation.

Sincerely,

Charles R M Bangham, ScD FRS

Academic Editor

PLOS Pathogens

Richard Koup

Section Editor

PLOS Pathogens

Kasturi Haldar

Editor-in-Chief

PLOS Pathogens

orcid.org/0000-0001-5065-158X

Michael Malim

Editor-in-Chief

PLOS Pathogens

orcid.org/0000-0002-7699-2064

The reviewers agreed that your findings are of interest and potential significance in the evolution of the ALV-J strain. However, reviewer 1 raises significant concerns on certain fundamental aspects of the experimental design, in particular the MOI used: these points need to be addressed before a revised manuscript can be considered.

Reviewer's Responses to Questions

**Part I - Summary**

Reviewer #1: In the present manuscript “N123I mutation in the ALV-J receptor-binding domain region enhances the viral replication ability by increasing binding affinity with chNHE1“, Yu et al. analyzed the effect of a single amino-acid substitution in the ALV-J gp85 on in vitro virus replication and receptor binding. The authors compared the prototype ALV-J isolate HPRS103 and the recent Chinese isolate JL08CH3-1 and mapped the difference to GP85. Subsequently, they reproduced the difference by substituting N123I within the HPRS103 envelope. Yu et al. attributed this to the rapid ALV-J radiation in China and compare it with SARS-CoV-2 evolution in recent years. Overall, the main finding of the study is correct, but should be better supported by experiments and its relevance for the virus evolution should be clearly explained.

The N123I is prevalently present in recent Chinese isolates from laying chicken breeds, but also in isolates from broilers, both in British isolates closely related to HPRS103 and in highly pathogenic American isolates. Thus, the N123I does not explain either the host range extension of the virus from broilers to egg-type chickens or the recent gp85 variation in Asia. Much of the background information and discussion is, therefore, irrelevant and should be deleted or replaced by a more plausible context.

I also suggest (in accordance to the aforementioned conceptual flaw) that Fig. 7 is put into Results and commented in a new chapter.

Reviewer #2: In this study, Yu and colleagues have found a strong linkage between the heightened replication efficacy of the layer strain JL08CH3-1 and the enhanced affinity exhibited by gp85 towards receptor chNHE1 compared to the prototype strain HPRS103. Particularly, the N123I mutation within the RBD region of gp85 precipitates an elevation in the binding affinity between gp85 and the receptor. They further demonstrated that N123I increased the binding capacity between gp85 and chNHE1, likely through enhancing the stability of gp85, expanding the interaction interface, and increasing the number of hydrogen bonds at the interaction interface. Furthermore, they systematically analyzed the amino acid sequence of ALV-J and conducted animal experiments, revealing that the N123I mutation enhanced the replication ability of ALV-J, making it more easily transmitted and likely to cause large epidemics. In general, these systematic studies are well-designed and the results make a valuable contribution to our understanding of the importance of amino acid mutations within the RBD region of the ALV-J gp85 protein in the viral evolution process. However, the manuscript should be improved before its acceptance for publication.

**Part II – Major Issues: Key Experiments Required for Acceptance**

Reviewer #1: 1. The virus replication in vitro was tested by virus titration, but the infection was performed at an extremely low dose of the virus (MOI 0.01). Under such conditions, the experiment can be easily biased by inaccurately measured titer of the virus stock (two virus variants with potentially different replication capacity). Virus replication should be quantified in parallel using an RT assay.

2. The binding capacity of JL08CH3-1 gp85 to chNHE1 was tested using protein-cell binding assay. However, virus-cell binding assay would be more direct and biologically relevant.

Reviewer #2: 1. Why did you choose the layer strain JL08CH3-1 as the representative strain for this study?

2. Why do you directly analyze the impact of gp85 on replication when you find that the replication ability of the broiler strain is stronger than that of the prototype strain？

**Part III – Minor Issues: Editorial and Data Presentation Modifications**

Reviewer #1: Minor points

1. Abstract, line 19; Introduction, line53. ALV-J is not present worldwide. Europe and the United states are ALV-J-free now.

2. Abstract, line 26,27. It should be consistently said that structural modeling as performed in this study is merely indicative and the results should be evaluated carefully.

3. Introduction is rather long and trivial textbook facts (such as lines 70-72) could be deleted. Reference 23 does not document the statement before.

4. In the chapter Binding capacity of JL08CH3-1...., the terms binding capacity, binding capability, binding rate, and binding ability are confused (same also in Fig. 2). Overall, the chapter is poorly written with a lot of unnecessary detail.

5. Results, line 158. The reference 36 seems to be incorrect at this place.

6. Discussion, lines 310-313. These two sentences do not make good sense.

7. Methods, line 390. Concentration of the cell extract with gp85 should be given.

8. Fig. 3B. Description of gp85 mutants does not correspond to the text and the alignment 3A.

Reviewer #2: 3. Line 62, replace 'show' with ' shows'.

4. Line 85, It is necessary to add abbreviations for simian-human immunodeficiency virus and feline panleukopenia virus, such as severe acute respiratory syndrome coronavirus 2 (SARS-CoV-2)

5. Line 105 and 106, replace ' log10 ' with 'log10 '.

6. How do you get the binding free energies of HPRS103-gp85complexed with TM-chEC1（ -300 kJ/mol） or N123I-gp85 complexed with TM-chEC1（-600 kJ/mol）,which is an average or a numerical value at a certain point?

7. Line 225, replace ' enhanced ' with ' enhances '.

8. Line 241, replace ' ALV-gp85 ' with ' ALV-J gp85 '.

9. Line 413, the Reed and Muench method, reference should be cited.

10. Structural analysis in material methods requires reference to support the credibility of the prediction method.

11. The labeling of the catalog numbers, companies, and origins of reagents and instruments within the Materials and Methods section is inconsistent.

12. Fig1A,1D, and 6D replacing ' LgTCID50/ml' with ' Log10TCID50/ml ' to be consistent with the description in the article. The '（）' of Fig1A should not be bolded.

13. Fig 1C and 6C legend, Scale bar needs to be added.

14. The format of the reference is not uniform, such as the format of author name (ref 1 VS ref 1), which needs to be modified according to the Author guide.

15. The manuscript requires extensive revision for grammar and style issues.

PLOS authors have the option to publish the peer review history of their article (what does this mean?). If published, this will include your full peer review and any attached files.

Reviewer #1: No

Reviewer #2: No
---

## [Decision Letter · Decision Letter 1]

20 Nov 2023

Dear Prof. Gao,

Thank you very much for submitting your manuscript "N123I mutation in the ALV-J receptor-binding domain region enhances the viral replication ability by increasing binding affinity with chNHE1" for consideration at PLOS Pathogens. As with all papers reviewed by the journal, your manuscript was reviewed by members of the editorial board and by several independent reviewers. The reviewers appreciated the attention to an important topic. Based on the reviews, we are likely to accept this manuscript for publication, providing that you modify the manuscript according to the review recommendations.

The reviewers agreed that the revised manuscript is greatly improved. However, reviewer 1 has made some comments and requirements for further minor changes to the manuscript that require your attention.

Sincerely,

Charles R M Bangham, ScD FRS

Academic Editor

PLOS Pathogens

Richard Koup

Section Editor

PLOS Pathogens

Kasturi Haldar

Editor-in-Chief

PLOS Pathogens

orcid.org/0000-0001-5065-158X

Michael Malim

Editor-in-Chief

PLOS Pathogens

orcid.org/0000-0002-7699-2064

The reviewers agreed that the revised manuscript is greatly improved. However, reviewer 1 has made some comments and requirements for further minor changes to the manuscript that require your attention.

Reviewer Comments (if any, and for reference):

Reviewer's Responses to Questions

**Part I - Summary**

Reviewer #1: In the present manuscript “N123I mutation in the ALV-J receptor-binding domain region enhances the viral replication ability by increasing binding affinity with chNHE1“, Yu et al. analyzed the effect of a single amino-acid substitution in the ALV-J gp85 on in vitro virus replication and receptor binding. The authors compared the prototype ALV-J isolate HPRS103 and the recent Chinese isolate JL08CH3-1 and mapped the difference to GP85. Subsequently, they reproduced the difference by substituting N123I within the HPRS103 envelope.

In the revised version, the authors have substantially improved the manuscript PPATHOGENS-D-23-01396R1. However, some of my criticisms and reservations have not been fully addressed and I suggest another revision of the manuscript.

Overall, the main finding of the study is correct, but some partial conclusions should be better supported by experiments with a more direct biological relevance.

Reviewer #2: After revision, the manuscript has been greatly improved. We believe that it now meets the journal's requirements and is ready for publication.

**Part II – Major Issues: Key Experiments Required for Acceptance**

Reviewer #1: I recommend that the binding capacity of JL08CH3-1 gp85 to chNHE1 should be tested using virus-cell binding assay in addition to the protein-cell binding assay. This experiment is not demanding and adds the biological relevance to the results. The protein-cell binding assay does not apply the gp85 in trimeric form and the results of both assays usually differ.

Reviewer #2: (No Response)

**Part III – Minor Issues: Editorial and Data Presentation Modifications**

Reviewer #1: 1. In introduction, the sentence „Epidemiological investigations have shown...“ (lines 57-58) does not make good sense.

2. Although the authors have modified the discussion slightly, it is still too speculative and unnecessarily long. I recommend toning down the discussion of structural implications of N123I substitution as it is based only on computative modelling, not actual structural analysis. I also advice to shorten the comparison of N123I with mutations increasing the receptor binding capacity of other viruses such as EboV and SARS-CoV-2.

3. In the discussion, I concern two sentences about the occurrence of N123I in American and Chinese strains of ALV-J (lines 299 to 303). The discussion gives the impression that these are two independent cases; can this be supported by phylogenetic data?

4. Although the authors declare proofreading by an English-speaking professional, I believe that the manuscript could have been smoother by extensive stylistic and argumentative editing.

Reviewer #2: (No Response)

PLOS authors have the option to publish the peer review history of their article (what does this mean?). If published, this will include your full peer review and any attached files.

Reviewer #1: No

Reviewer #2: No

Figure Files:

Data Requirements:

Reproducibility:

References:

---

## [Editor Report · Decision Letter 2]

28 Dec 2023

Dear Prof. Gao,

We are pleased to inform you that your manuscript 'N123I mutation in the ALV-J receptor-binding domain region enhances the viral replication ability by increasing binding affinity with chNHE1' has been provisionally accepted for publication in PLOS Pathogens.

Best regards,

Charles R M Bangham, ScD FRS

Academic Editor

PLOS Pathogens

Richard Koup

Section Editor

PLOS Pathogens

Kasturi Haldar

Editor-in-Chief

PLOS Pathogens

orcid.org/0000-0001-5065-158X

Michael Malim

Editor-in-Chief

PLOS Pathogens

orcid.org/0000-0002-7699-2064
---

## [Editor Report · Acceptance letter]

1 Feb 2024

Dear Prof. Gao,

We are delighted to inform you that your manuscript, "N123I mutation in the ALV-J receptor-binding domain region enhances the viral replication ability by increasing binding affinity with chNHE1," has been formally accepted for publication in PLOS Pathogens.

Best regards,

Michael Malim

Editor-in-Chief

PLOS Pathogens

orcid.org/0000-0002-7699-2064